# Vanillin Induces Relaxation in Rat Mesenteric Resistance Arteries by Inhibiting Extracellular Ca^2+^ Influx

**DOI:** 10.3390/molecules28010288

**Published:** 2022-12-29

**Authors:** Sooyeon Choi, Chae Eun Haam, Eun-Yi Oh, Seonhee Byeon, Soo-Kyoung Choi, Young-Ho Lee

**Affiliations:** Department of Physiology, Yonsei University College of Medicine, 50 Yonseiro, Seodaemun-gu, Seoul 03722, Republic of Korea

**Keywords:** vanillin, Ca^2+^, mesenteric resistance arteries, relaxation, vasodilation

## Abstract

Vanillin is a phenolic aldehyde, which is found in plant species of the *Vanilla* genus. Although recent studies have suggested that vanillin has various beneficial properties, the effect of vanillin on blood vessels has not been studied well. In the present study, we investigated whether vanillin has vascular effects in rat mesenteric resistance arteries. To examine the vascular effect of vanillin, we measured the isometric tension of arteries using a multi-wire myograph system. After the arteries were pre-contracted with high K^+^ (70 mM) or phenylephrine (5 µM), vanillin was administered. Vanillin induced concentration-dependent vasodilation. Endothelial denudation or treatment of eNOS inhibitor (L-NNA, 300 μM) did not affect the vasodilation induced by vanillin. Treatment of K^+^ channel inhibitor (TEA, 10 mM) or sGC inhibitor (ODQ, 10 μM) or COX-2 inhibitor (indomethacin, 10 μM) did not affect the vanillin-induced vasodilation either. The treatment of vanillin decreased the contractile responses induced by Ca^2+^ addition. Furthermore, vanillin significantly reduced vascular contraction induced by BAY K 8644 (30 nM). Vanillin induced concentration-dependent vascular relaxation in rat mesenteric resistance arteries, which was endothelium-independent. Inhibition of extracellular Ca^2+^ influx was involved in vanillin-induced vasodilation. Treatment of vanillin reduced phopsho-MLC_20_ in vascular smooth muscle cells. These results suggest the possibility of vanillin as a potent vasodilatory molecule.

## 1. Introduction

The cardiovascular system comprises the heart, veins, arteries, and capillaries. The arteries carry blood from the heart to the tissues to deliver oxygen and nutrients. Resistance arteries are blood vessels with an inner diameter of <400 μm that are known to play an important role in generating peripheral resistance [1]. These blood vessels include small arteries and arterioles. According to Poiseuille’s law, resistance of the arteries is inversely proportional to the radius to the fourth power [2]. Thus, a small decrease in the lumen diameter of resistance arteries and arterioles markedly increases peripheral resistance. It is well known that essential hypertension is associated with elevated peripheral resistance [3]. The causes for increased peripheral resistance could be functional, mechanical, or structural abnormalities in resistance arteries. The functional abnormalities include increased vascular tone due to augmented vasoconstriction and/or impaired vasodilation. Vasodilators act to lower peripheral resistance by relaxing blood vessels, which results in a decrease in blood pressure. There are many anti-hypertensive drugs that produce vasodilation by acting directly on the smooth muscle of the arteries [4]. Although many anti-hypertensive medications have been widely used to treat and cure patients, their adverse effects remain a challenge. Therefore, it is an important task to find vasodilatory substances that can act stably without side effects to treat hypertensive patients.

Vanillin (4-hydroxy-3-methoxybenzaldehyde, C_8_H_8_O_3_) is a phenolic aldehyde, which is found in plant species of the Vanilla genus [5,6]. Vanillin is the main component of vanilla extract and has been commonly used as a flavoring agent or a food additive around the world [7]. Recently, several studies have suggested that vanillin has biological properties, such as antioxidant [8], anti-inflammatory [9], anti-tumor [10], anti-diabetic [11], and anti-angiogenic [12] properties. Interestingly, Raffai et al. reported that vanillin and vanillin analogues induced relaxation in porcine coronary and basilar arteries [13]. Furthermore, in a previous study, we have reported that *Trachelospermi caulis* extract induces vasodilation in rat mesenteric resistance arteries. In that study, we found that vanillin as an active substance can induce vasodilation [14]. However, despite the variety of studies, the effects of vanillin on the cardiovascular system have not been sufficiently studied. Therefore, in the present study, we investigated the direct vascular effect of vanillin in rat mesenteric resistance arteries and its underlying mechanism.

## 2. Results

### 2.1. Effect of Vanillin on the Contraction Induced by High K^+^ or Phenylephrine

To observe the vascular effect of vanillin, vanillin was cumulatively administered to mesenteric arteries pre-contracted with high K^+^ (70 mM) or phenylephrine (PE, 5 μM). Vanillin (0.03 mM–20 mM) induced vasodilation in the mesenteric resistance arteries in a concentration-dependent manner (Figure 1A,B). The maximal value of vanillin-induced vasodilation is 101.22 ± 1.21% in the arteries pre-contracted with high K^+^, and 102.25 ± 1.40% in the arteries pre-contracted with PE. The EC_50_ of vanillin is 1.07 ± 10.36 mM (Figure 1C).

### 2.2. Endothelium-Independent Vasodilation Induced by Vanillin

To define whether vanillin-induced vasodilation is dependent on the endothelium, vanillin was administered in endothelium-intact (Figure 2A) or endothelium-denuded (Figure 2B) mesenteric resistant arteries. The maximal value of vanillin-induced vasodilation is 101.07 ± 1.28% in the endothelium intact arteries, and 101.08 ± 0.54% in the endothelium-denuded arteries. There was no significant difference in the vasodilatory effect of vanillin between endothelium-intact and endothelium-denuded arteries (Figure 2C).

### 2.3. Effect of eNOS Inhibitor (L-NNA), sGC Inhibitor (ODQ), and COX Inhibitor (Indomethacin) on the Vasodilation Induced by Vanillin

To confirm that the endothelium is not involved in the vasodilatory effect of vanillin, and to further investigate whether the nitric oxide (NO)/cyclic guanosine monophosphate (cGMP) and cyclooxygenase (COX)/prostacyclin (PGI_2_) pathways are involved in vanillin-induced vasodilation, the arteries were incubated for 20 min with an endothelial nitric oxide (eNOS) inhibitor, Nω-Nitro-L-arginine (L-NNA, 300 µM), or a soluble guanylyl cyclase (sGC) inhibitor, 1H-(1,2,4)oxadiazolo [4,3-a]quinoxalin-1-one (ODQ, 10 µM), or a COX inhibitor, indomethacin (10 µM), before being contracted with PE (5 µM). The vasodilatory responses induced by vanillin were 99.97 ± 2.96%, 102.06 ± 1.81%, 102.26 ± 1.94% in the presence of eNOS inhibitor (L-NNA), sGC inhibitor (ODQ), and COX inhibitor (indomethacin), respectively (Figure 3).

### 2.4. Effect of TEA on the Vasodilation Induced by Vanillin

To define whether the K^+^ channel is involved in vanillin-induced vasodilation, the non-specific K^+^ channel blocker, tetraethylammonium (TEA, 10 mM), was pre-treated 20 min before being contracted with PE (5 µM). Treatment of TEA did not alter the vasodilatory response of vanillin. The vasodilatory responses induced by vanillin in the presence of TEA were 101.6 ± 1.15% (Figure 4).

### 2.5. Effect of Vanillin on Extracellular Ca^2+^-Induced Vascular Contraction

To investigate whether the vasodilatory effect of vanillin is associated with the inhibition of extracellular Ca^2+^ influx, we examined the contraction responses caused by the addition of CaCl_2_ (0.1–2.0 mM) in the arteries incubated in a Ca^2+^-free K-H solution containing a sarcoplasmic reticulum Ca^2+^-ATPase (SERCA) inhibitor, cyclopiazonic acid (CPA, 5 µM), and KCl (70 mM). It was confirmed that the contraction responses caused by the repeated addition of CaCl_2_ were not changed (Figure 5A). Pre-treatment of vanillin (3 mM) significantly reduced the contractile responses induced by the cumulative addition of CaCl_2_ (Figure 5B,C).

### 2.6. Effect of Vanillin on the BAY K 8644-Induced Contraction

To confirm that inhibition of extracellular Ca^2+^ influx is involved in the vanillin-induced relaxation, the arteries were contracted with an L-type voltage-gated calcium channel (VGCC) activator, BAY K 8644, and then vanillin was administered in the mesenteric arteries. Before being contracted, the arteries were incubated with a K-H solution containing 15 mM of K^+^ to create an environment where VGCC could be opened. Administration of the vehicle (K-H solution, 3 µL–2090 µL) did not alter BAY K 8644-induced vascular contraction (Figure 6A). Vanillin relaxed mesenteric arteries contracted by BAY K 8644 in a concentration-dependent manner. (Figure 6B,C).

### 2.7. Decreased Phosphorylation of MLC_20_ by Vanillin in Vascular Smooth Muscle Cells

To investigate whether vanillin-induced relaxation was caused by decreased phosphorylation of MLC_20_, the phosphorylation and expression levels of MLC_20_ were measured in vascular smooth muscle cells (VSMCs, Figure 7). The administration of phenylephrine (5 µM) increased the phosphorylation level of MLC_20_ in VSMCs, which was reduced by the treatment with vanillin (3 mM).

## 3. Discussion

Although vanillin is the most popular flavor compound in the world and its biological properties are well reported, its effect on the cardiovascular system is not sufficiently studied. Moreover, in a previous study [14], we found that *Trachelospermi caulis* extract, a traditional herbal medicine, exhibited a significant vasodilating effect; further, vanillin might be a critical active component inducing the effect. However, a detailed study on the vasorelaxant effect of vanillin and its mechanism of action was indispensable. Thus, the purpose of this study was to investigate the direct vascular effect of vanillin in rat mesenteric resistance arteries and to determine the underlying mechanism. We demonstrated that vanillin induced vasodilation in arteries contracted with KCl or PE in a concentration-dependent manner. Furthermore, endothelial removal, and the pre-treatment of an eNOS inhibitor (L-NNA), sGC inhibitor (ODQ), and COX inhibitor (indomethacin) did not alter the vasodilatory effect of vanillin. The non-specific K^+^ channel blocker, TEA, did not affect the vanillin-induced vascular relaxation either. Vanillin significantly inhibited Ca^2+^-induced contraction, which was confirmed by the inhibition of BAY K 8644-induced contraction.

In the present study, the vasorelaxant effect of vanillin was observed in rat mesenteric resistance arteries contracted with high K^+^ or PE (Figure 1). These vasocontractile substances were used to determine whether vanillin could induce vasodilation in the arteries contracted via direct membrane depolarization and/or via agonist stimulation. We did not observe a significant difference in vanillin-induced vasodilation in both cases, which means vanillin induces a relaxation response regardless of the type of stimulus.

Vasodilation occurs when the smooth muscle cells relax in the blood vessel walls. Relaxation of the smooth muscle cells could be due to either removal of a contractile stimulus or the direct action of vasodilatory substances [15]. The vascular endothelium is a monolayer of endothelial cells lining the luminal surface of vessels [16]. In response to various stimuli, the endothelium releases vasodilatory substances, such as NO, prostacyclin (prostaglandin I_2_; PGI_2_), endothelium derived hyperpolarizing factor (EDHF), and vasocontractile substances such as thromboxane (TXA_2_) and endothelin-1 (ET-1) [17]. In the present study, we examined whether vanillin induces vasodilation via the endothelium (Figure 2). We found that the removal of the endothelium did not alter the vasodilatory effect of vanillin in rat mesenteric resistance arteries. In order to confirm that vanillin-induced vasodilation is endothelium-independent, we used the eNOS inhibitor, L-NNA (Figure 3A). Treatment with L-NNA did not affect vanillin-induced vasodilation in mesenteric arteries. Nitric oxide is generated by eNOS in the endothelium, then it diffuses into smooth muscle cells and activates sGC to increase intracellular cyclic guanosine monophosphate (cGMP) concentration, which leads to relaxation [18]. We found that the sGC inhibitor, ODQ, did not alter vanillin-induced vasodilation (Figure 3B). Our results show that vanillin does not induce relaxation of the blood vessels through the NO/cGMP pathway. These findings are in accordance with a previous study by Raffai et al. that showed vanillin induced endothelium-independent vasodilation in porcine coronary and basilar arteries [13]. They found that endothelial removal and treatment with the eNOS inhibitor, L-NAME, did not affect vanillin-induced vasodilation. In the present study, we also investigated whether another vasodilatory substance, PGI_2_, is involved in vanillin-induced vascular relaxation (Figure 3C). It is known that PGI_2_ is produced by COX [19]. We found that pre-treatment with indomethacin, a non-selective COX inhibitor, did not affect vanillin-induced vasodilation. Taken together, vanillin induces vasodilation via an endothelium-independent mechanism.

After we confirmed that vanillin-induced vasodilation is endothelium-independent, we examined whether vanillin directly acts on smooth muscle cells to induce relaxation. Reduction of extracellular Ca^2+^ influx or Ca^2+^ release from intracellular Ca^2+^ stores (sarcoplasmic reticulum, SR) results in the relaxation of smooth muscle cells [15]. The K^+^ channels determine and regulate the membrane potential of vascular smooth muscle cells [20]. Regulation of the membrane potential through these channels affects the open-state probability of voltage-gated Ca^2+^ channels (VGCCs), which can induce Ca^2+^ influx, resulting in smooth muscle contraction [21]. In contrast, membrane hyperpolarization contributes to the closure of VGCCs to block the influx of extracellular Ca^2+^, leading to relaxation of the smooth muscle cells. In the present study, we treated mesenteric arteries with the non-specific K^+^ channel blocker, TEA, to examine the involvement of the K^+^ channel in vanillin-induced vasodilation (Figure 4). It is known that TEA blocks all types of K^+^ channels [22]. Pre-treatment with TEA did not alter the vasodilatory effect of vanillin, which indicates that the vasodilatory effect of vanillin is not associated with the activation of K^+^ channels.

Since we found that the endothelium and K^+^ channel are not involved in vanillin-induced vasodilation, next, we examined whether vanillin directly inhibits the increase in cytosolic Ca^2+^ in smooth muscle cells. We did not administer Ca^2+^ channel blockers to the arteries because blockage of the Ca^2+^ channels did not induce enough contraction to test the effect of vanillin. Then, we alternatively tested the effect of vanillin (Figure 5). The mesenteric arteries were incubated in a Ca^2+^ free K-H solution containing CPA to remove intracellular free Ca^2+^. Then, 70 mM K^+^ was administered to enable the opening of the VGCCs. The addition of Ca^2+^ induced vasoconstriction, which was reduced by pre-treatment with vanillin. In order to confirm that vanillin directly acts on the contraction induced by VGCC activation, we used the VGCC activator, BAY K 8644. We found that vanillin induced relaxation in a concentration-dependent manner in mesenteric resistance arteries pre-contracted with BAY K 8644 (Figure 6). From these results, we assumed that extracellular Ca^2+^ influx was inhibited by the treatment with vanillin.

In vascular smooth muscle cells, an increase in intracellular Ca^2+^ concentration leads to activation of myosin light chain kinase (MLCK), which phosphorylates MLC_20_. Phosphorylated MLC_20_ evokes the activation of myosin ATPase, which results in contraction [23]. Thus, the phosphorylation of MLC_20_ is essential for smooth muscle contraction. In contrast, a decrease in intracellular Ca^2+^ allows the opposite reaction. Since we observed that vanillin reduced Ca^2+^-induced contraction in mesenteric arteries, we investigated whether vanillin affects the phosphorylation level of MLC_20_ in vascular smooth muscle cells. We found that treatment with PE increased the phosphorylation level of MLC_20_. Interestingly, co-treatment with PE and vanillin significantly reduced the phosphorylation level of MLC_20_ compared to treatment with PE alone. Although we did not directly measure the changes in intracellular Ca^2+^ concentration induced by vanillin, the above results suggest that vanillin reduces the phosphorylation of MLC_20_, and ultimately, causes vascular relaxation.

In the present study, we suggest vanillin as a potent vasodilating substance since it exhibits a significant relaxing effect. Future in vivo studies are needed to show the potential therapeutic effect of vanillin to treat high blood pressure.

## 4. Materials and Methods

### 4.1. Animals and Tissue Preparation

Ten to twelve week old male Sprague Dawley rats were used in this study. After the rats were sacrificed, the mesenteric resistance arteries were extracted and placed in an ice-cold Krebs–Henseleit (K-H) solution (composition (mM): NaCl 119, NaHCO_3_ 25, glucose 11.1, KCl 4.6, MgSO_4_ 1.2, KH_2_PO_4_ 1.2, and CaCl_2_ 2.5) aerated with 95% O_2_ and 5% CO_2_. The adipose and connective tissues surrounding the mesenteric arteries were removed using forceps under the microscope (model SZ-40, Olympus, Shinjuku-ku, Tokyo, Japan). The second branches of the mesenteric arteries (diameter of 250–300 μm) were cut into sections 2–3 mm long and used in this study. The endothelium was removed by gently rubbing, using forceps if necessary.

### 4.2. Measurement of Isometric Tension in Mesenteric Arteries

The mesenteric artery segments were mounted in a wire myograph (model 620M, Danish Myotechnology, Aarhus, Denmark) for measuring the isometric tension. Arterial rings were bathed in 37 °C K-H solution, aerated with 95% O_2_ and 5% CO_2_. Arteries were equilibrated for 20 min and stretched to their optimal resting tension (4 mN). The vascular responses were measured by contracting the arteries to KCl (70 mM) or phenylephrine (PE, 5 μM), followed by a cumulative addition of vanillin (0.03–20 mM). To investigate the vasodilatory mechanism of vanillin, eNOS inhibitor (L-NNA) or sGC inhibitor (ODQ) or COX inhibitor (indomethacin) or TEA were pre-treated for 20 min, and then the relaxation response of the vanillin was measured. To determine the involvement of Ca^2+^ influx in vanillin-induced relaxation, the normal K-H solution was replaced with a Ca^2+^ free K-H solution containing 70 mM of KCl and cyclopiazonic acid (CPA, 5 μM), then CaCl_2_ was cumulatively added. Vanillin (3 mM) was used to treat the same arteries 20 min before another cumulative addition of CaCl_2_. The CaCl_2_-induced contraction was calculated as the percentage of maximum contraction induced by KCl (70 mM). Some arteries were pre-contracted by BAY K 8644 (30 nM) in K-H solution containing 15 mM KCl to investigate vanillin-induced vasodilation.

### 4.3. Isolation and Culture of Vascular Smooth Muscle Cells

After the rats were sacrificed, the aortas were excised, the fat and connective tissues were removed. The aortas were cut into small segments and transferred into a tube containing collagenase (1 mg/mL, Worthington Biomedical Corporation, Lakewood Township, NJ, USA) and elastase (0.5 mg/mL, Calbiochem, San Diego, CA, USA) in Dulbecco’s Modified Eagle Medium (DMEM, Gibco, Waltham, MS, USA) at 37 °C for 30 min. After trituration and centrifugation, the cells were seeded in culture dishes (Corning, New York, NY, USA) and cultivated in DMEM supplemented with 10% FBS, 100 IU/mL penicillin, and 10,000 µg/mL streptomycin (Gibco) at 37 °C, in 5% CO_2_. The early passage cells (between two and four) were used in this study.

### 4.4. Western Blot Analysis

The cultured VSMCs were treated with a vehicle (0.05% DMSO) or PE (5 µM) or PE (5 µM) with vanillin (3 mM), and then homogenized in an ice-cold lysis buffer, as described previously [24]. Western blot analysis was performed for the total MLC_20_ and phosphorylated MLC_20_ (1:1000 dilution; Cell Signaling, Boston, MA, USA). Blots were stripped and then reprobed with the β-actin antibody (1:2000 dilution; Santa Cruz Biotechnology, Santa Cruz, CA, USA) to verify the equal loading between the samples.

### 4.5. Chemicals and Reagents

Phenylephrine, ACh, eNOS inhibitor (L-NNA), sGC inhibitor (ODQ), TEA, and vanillin were purchased from Sigma-Aldrich (St. Louis, MO, USA). COX inhibitor (Indomethacin) was obtained from Calbiochem (Darmstadt, Germany). CPA was obtained from Enzo Life Sciences (Farmingdale, NY, USA).

### 4.6. Statistical Analysis

All values are expressed as mean ± standard deviations. A one-way or two-way ANOVA was used to compare the means of the comparison groups when appropriate. Comparisons between groups were performed with *t*-tests when the ANOVA test was statistically significant. Values of * *p* < 0.05 were considered statistically significant.

## 5. Conclusions

In the present study, we discovered that vanillin induced vascular relaxation in rat mesenteric resistance arteries in a concentration-dependent manner. We also found that the endothelium, NO/cGMP pathway, and prostacyclin are not involved in vanillin-induced vasodilation. The inhibition of extracellular Ca^2+^ influx was associated with vanillin-induced vasodilation. Our data suggest that vanillin has the therapeutic potential to treat high blood pressure.

## Figures and Tables

**Figure 1 molecules-28-00288-f001:**
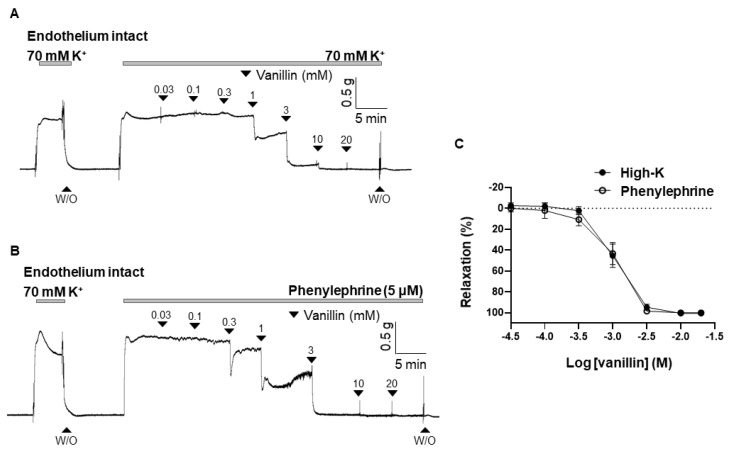
Vanillin-induced vasodilation in rat mesenteric arteries. Representative traces showing responses to cumulative administration of vanillin (0.03–20 mM) on 70 mM-K^+^ (**A**) or phenylephrine (5 µM)-induced contraction (**B**). Statistical analysis of the relaxation response to vanillin (**C**). Mean ± SD (*n* = 7). (W/O: wash out).

**Figure 2 molecules-28-00288-f002:**
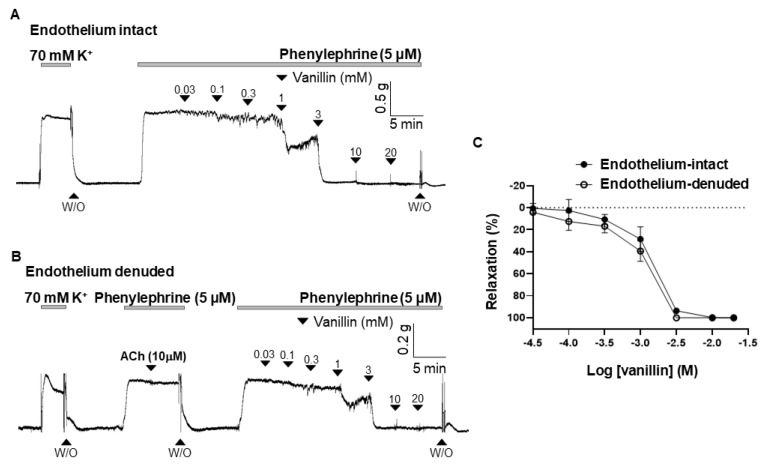
Endothelium-independent vasodilation induced by vanillin. Representative trace showing vanillin-induced vasodilation in the endothelium-intact (**A**) and endothelium-denuded mesenteric arteries (**B**). Statistical analysis of vanillin-induced vasodilation (**C**). Mean ± SD (*n* = 5). (ACh: acetylcholine; W/O: wash out).

**Figure 3 molecules-28-00288-f003:**
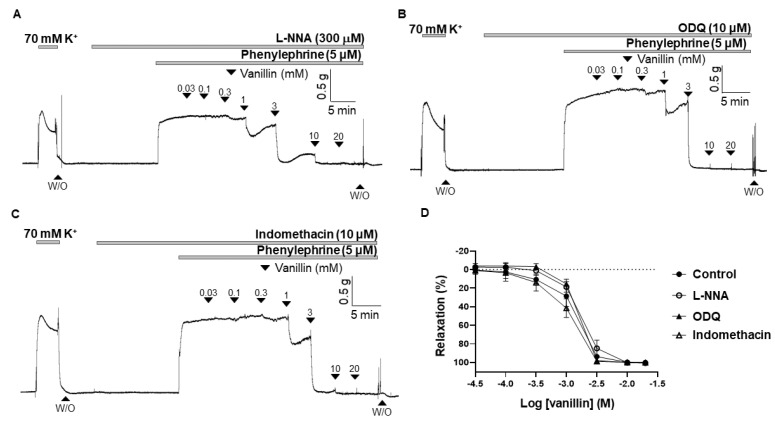
Effect of eNOS inhibitor (L-NNA), sGC inhibitor (ODQ), and COX inhibitor (indomethacin) on vanillin-induced vasodilation. Representative trace showing vanillin-induced vascular relaxation in the presence of L-NNA (**A**), ODQ (**B**), and indomethacin (**C**). Statistical analysis of the vasodilation in response to vanillin in the presence of L-NNA, ODQ and indomethacin (**D**). Relaxation is expressed as the percentage of the contraction induced by PE (5 µM). Mean ± SD (*n* = 5). (L-NNA: Nω-Nitro-L-arginine; ODQ: 1H-[1,2,4]-oxadiazolo-[4,3-α]-quinoxalin-1-one; W/O: wash out).

**Figure 4 molecules-28-00288-f004:**
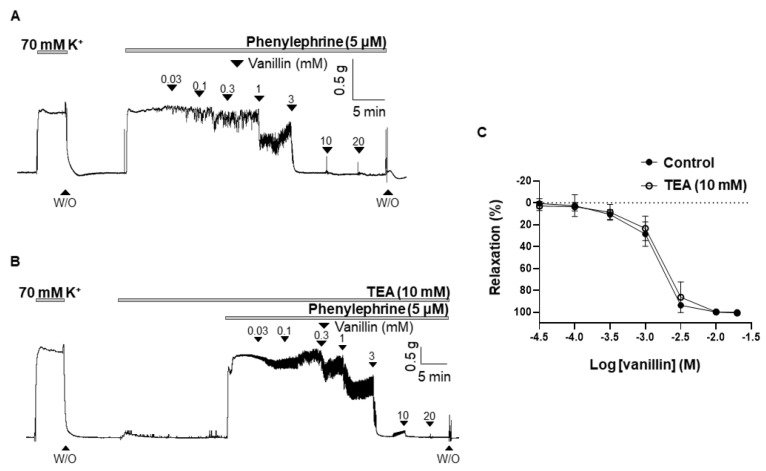
Effect of a K^+^ channel blocker on vanillin-induced vasodilation. Effects of vanillin on the pre-contracted arteries with PE (5 µM) in the absence of TEA (**A**) and in the presence of TEA (**B**). Statistical analysis of the relaxation response of vanillin in the presence of TEA (**C**). Relaxation of arteries is expressed as the percentage of the contraction induced by PE (5 µM). Mean ± SD (*n* = 7). (TEA: tetraethylammonium; W/O: wash out).

**Figure 5 molecules-28-00288-f005:**
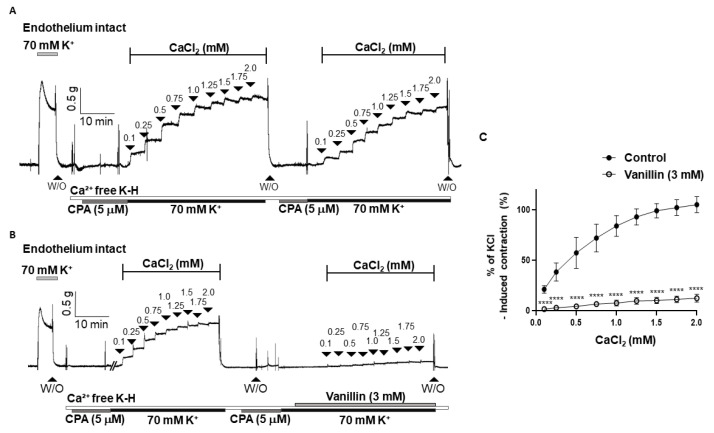
Vanillin inhibited extracellular Ca^2+^-induced vascular contraction. Representative traces showing the contraction responses by repeated addition of Ca^2+^ (0.1–2.0 mM) in the absence of vanillin (3 mM, (**A**)) and in the presence of vanillin (3 mM, (**B**)). Statistical analysis of contraction induced by CaCl_2_ in the mesenteric arteries with or without vanillin (**C**). Mean ± SD (*n* = 7). **** *p* < 0.0001 (W/O: wash out; CPA: cyclopiazonic acid).

**Figure 6 molecules-28-00288-f006:**
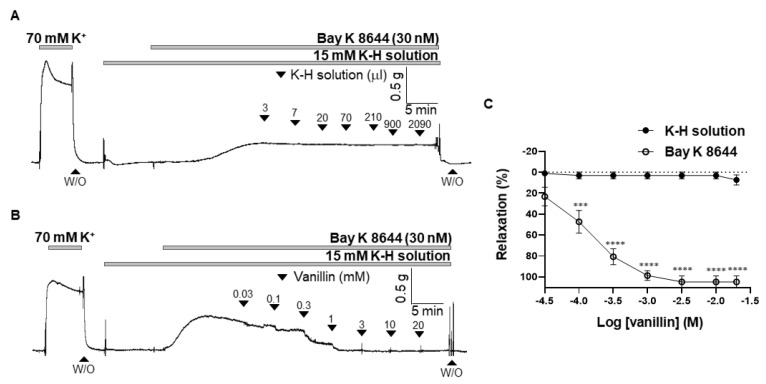
Vanillin reduced contraction induced by L-type voltage-gated calcium channel activation. Representative trace showing the effect of the vehicle (K-H solution, (**A**)) and vanillin (**B**) in the BAY K 8644-induced contraction. Statistical analysis of relaxation induced by vanillin in the mesenteric arteries pre-constricted by BAY K 8644 (**C**). Mean ± SD (*n* = 7). *** *p* < 0.001, **** *p* < 0.0001 (W/O: wash out).

**Figure 7 molecules-28-00288-f007:**
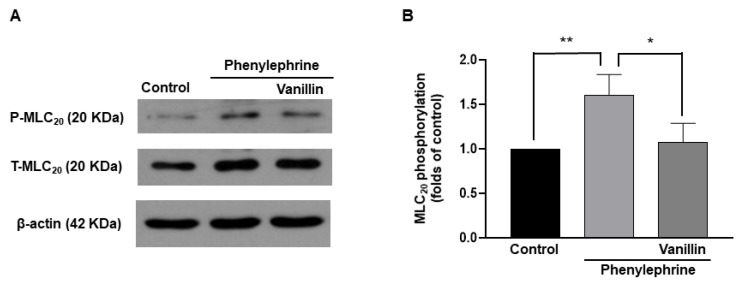
Effect of vanillin on the phosphorylation of MLC_20_. Representative Western blot analysis for phosphorylated MLC_20_ (P–MLC_20_) and total MLC_20_ (T–MLC_20_) in control VSMCs, VSMCs treated with phenylephrine (5 µM), and VSMCs co-treated with phenylephrine (5 µM) and vanillin (3 mM), (**A**). Quantitative data for phosphorylated MLC_20_ (**B**). *n* = 4, * *p* < 0.05, ** *p* < 0.005.

## Data Availability

Not applicable.

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
