# Peer review of "Vanillin Induces Relaxation in Rat Mesenteric Resistance Arteries by Inhibiting Extracellular Ca2+ Influx"

_molecules, 2022, doi:10.3390/molecules28010288_

Round 1

Reviewer 1 Report

Please delete the sentence from the previous revision:

Authors should discuss the results and how they can be interpreted from the perspective of previous studies and of the working hypotheses. The findings and their implications should be discussed in the broadest context possible. Future research directions may also be high- lighted.

And I do agree with that previous revision. Too much of mechanism is included rather then future perspectives. If you do perform a feeding experimentation this would be of great benefit.

Its better to use COX inhibitor rather then indomethacin, the same for ODQ sGC inhibitor and others through the text including the abstract.

Also western bloots describing eNOS and COX will be of great importance to this manuscript rather than plain isometric studies.

Author Response

We thank the editor and the reviewers for their careful review and do appreciate the constructive comments that strengthen our manuscript. The manuscript was revised in accordance with the suggestions of the reviewers.  

List of Changes:

  1. Figure 5 was changed. (Concentration of vanillin was added)
  2. Indomethacin and ODQ were changed to COX inhibitor and sGC inhibitor, respectively. (Page 3 and 4)
  3. A reference was added. (references 14)
  4. Title of the manuscript was changed.
  5. Discussion section was changed. (Page 7 and 8)

Responses to Reviewers:

Reviewer 1

Comment 1: Authors should discuss the results and how they can be interpreted from the perspective of previous studies and of the working hypotheses. The findings and their implications should be discussed in the broadest context possible. Future research directions may also be high-lighted.

Answer: We thank the reviewer for raising this concern. In the previous study, we reported that Trachelospermi caulis extract induces vasodilation in rat mesenteric resistance arteries. In that study, we found that vanillin as an active substance to induce vasodilation. It is very important to find substances to treat or prevent cardiovascular diseases such as hypertension. In this respect, vanillin is expected to be a good target substance. According to the reviewer’s opinion, we added this in the revised manuscript. (Please see page 7 and 8)

Comment 2: And I do agree with that previous revision. Too much of mechanism is included rather than future perspectives. If you do perform a feeding experimentation this would be of great benefit.

Answer: We thank the reviewer for raising this concern. According to the reviewer’s comment, we included future perspectives in the discussion. We also agree that a feeding experiment would be of great benefit. However, in the present study, we would like to suggest vanillin as a potential vasodilator through presenting the direct vascular effect. As the reviewer’s comment, we plan to perform in-depth in vivo experiment with vanillin for the future study. It takes longer experimental period to perform the experiment in vivo, and we only have ten days for the revision. We hope to show in vivo research data in the independent study in the near future. (Please see page 7 and 8)

Comment 3: It’s better to use COX inhibitor rather than indomethacin, the same for ODQ, sGC inhibitor and others through the text including the abstract.

Answer: We thank the reviewer for raising this concern. According to the reviewer’s comment, we changed indomethacin and ODQ to COX inhibitor and sGC inhibitor, respectively in the revised manuscript. (Please see page 3 and 4)

Comment 4: Also western blots describing eNOS and COX will be of great importance to this manuscript rather than plain isometric studies.

Answer: We thank the reviewer for raising this concern. However, since we did not observe any significant changes by eNOS inhibitor and COX inhibitor in the functional study, we think the western blot analysis for MLC20 is more appropriate for the research hypothesis.

Reviewer 2 Report

The present study reports the vasorelaxant effects of vanillin in rat resistance arteries. This effect is endothelium-independent, and involves modulation of Ca2+ influx and phosphorylation of myosin light chain kinase MCL20. Methodology was well designed and results adequately presented, as well as Conclusion opens for further investigation. The vasorelaxant effect of vanillin in resistance vessels was previously reported by the authors in Haam et al. (2022) as an identified compound from Trachelospermi cauli, but this paper was not cited. Therefore, considering few reports concerning vanillin-induced vasorelaxant effect, I strongly recommend the citation of this paper in order to create knowledge about this natural compound.

Moreover, the first complete report concerning these effect and mechanisms thereof is Raffai et al. (2015), from the deceased Prof. Vanhoute’s group. Then, one of the main criticism I do to the present work is that reference must be more exploited in order to fundament current novel findings. Thus, one of the major points that could achieve this view at a glance is to propose a title which came authors to novelty. At a first look, there is no novelty in “Vanillin induces vascular relaxation in rat mesenteric resistance arteries” due to Haam’s work. Therefore, I propose to complement the title with the mechanism underlying this effect in order to let it more attractive and novel.

As minor corrections, please follow:

- In Topic 2.5 and 4.2, please report the concentration of vanillin used in this protocol of extracellular Ca+-induced contraction, as well as in the Figure 5.

- In Topic 2.5 and 4.2, why authors have decided to assess only one concentration of vanillin? I suggest to perform CaCl2 curves in the present of at least three concentrations of vanillin. Otherwise, please justify why only one was used, as well as which one.

- The effect of vanillin on mobilization of intracellular calcium stores is also important to assess the role of vanilin in calcium homeostasis in artery preparations. I believe this approach might be considered. Did the authors have considered it? Please, clarify this question. 

Author Response

We thank the editor and the reviewers for their careful review and do appreciate the constructive comments that strengthen our manuscript. The manuscript was revised in accordance with the suggestions of the reviewers.  

Comment 1: The present study reports the vasorelaxant effects of vanillin in rat resistance arteries. This effect is endothelium-independent, and involves modulation of Ca2+ influx and phosphorylation of myosin light chain kinase MCL20. Methodology was well designed and results adequately presented, as well as Conclusion opens for further investigation. The vasorelaxant effect of vanillin in resistance vessels was previously reported by the authors in Haam et al. (2022) as an identified compound from Trachelospermi cauli, but this paper was not cited. Therefore, considering few reports concerning vanillin-induced vasorelaxant effect, I strongly recommend the citation of this paper in order to create knowledge about this natural compound.

Answer: We thank the reviewer for the constructive comment. In the previous study, we reported that Trachelospermi caulis extract induces vasodilation in rat mesenteric resistance arteries. In that study, we found that vanillin as an active substance to induce vasodilation. According to the reviewer’s opinion, we added this in the revised manuscript. (Please see page 2 and 7)

Comment 2: Moreover, the first complete report concerning these effect and mechanisms there of is Raffai et al. (2015), from the deceased Prof. Vanhoute’s group. Then, one of the main criticism I do to the present work is that reference must be more exploited in order to fundament current novel findings. Thus, one of the major points that could achieve this view at a glance is to propose a title which came authors to novelty. At a first look, there is no novelty in “Vanillin induces vascular relaxation in rat mesenteric resistance arteries” due to Haam’s work. Therefore, I propose to complement the title with the mechanism underlying this effect in order to let it more attractive and novel.

Answer: We thank the reviewer for the constructive comment. We totally agree with the reviewer. In the revised manuscript, we changed the title according to the reviewer’s opinion. (Vanillin induces relaxation in rat mesenteric resistance arteries by inhibiting extracellular Ca2+ influx.)

Comment 3:  In Topic 2.5 and 4.2, please report the concentration of vanillin used in this protocol of extracellular Ca2+-induced contraction, as well as in the Figure 5.

Answer: We apologize for the mistake. According to the reviewer’s comment, we added the description of vanillin concentration in the revised manuscript. (Please see page 5 and 9)

Comment 4: In Topic 2.5 and 4.2, why authors have decided to assess only one concentration of vanillin? I suggest to perform CaCl2 curves in the present of at least three concentrations of vanillin. Otherwise, please justify why only one was used, as well as which one.

Answer: We thank the reviewer for the comment. In order to show that vanillin relaxes mesenteric arteries through inhibition of Ca2+ influx, it would be good to show that cumulative addition of Ca2+ induces gradual and clear increase in contraction first and then pre-treatment of vanillin inhibits these responses. We thought that showing the results of the experiment at different concentrations of vanillin may confuse the reader and does not give clear explanation. We think the concentration of vanillin that induces maximum response would clearly show the effect of vanillin. Furthermore, in the later experiment with BAY K 8644, we showed concentration-dependent relaxation by vanillin, we think the one concentration of vanillin would be enough in the CaCl2 curves.

Comment 5:  The effect of vanillin on mobilization of intracellular calcium stores is also important to assess the role of vanillin in calcium homeostasis in artery preparations. I believe this approach might be considered. Did the authors have considered it? Please, clarify this question.

Answer: We thank the reviewer for the constructive comment. Vanillin-induced vasodilation might be due to the reduction of intracellular Ca2+ concentration through either blocking extracellular Ca2+ influx or storing Ca2+ to the SR. Although we could not exclude the possibility that vanillin induces mobilizing of Ca2+ to SR, we think vanillin-induced relaxation is due to the inhibition of extracellular Ca2+. Because we clearly showed that vanillin-induced vascular relaxation is significant and the contraction induced by extracellular Ca2+ is inhibited by vanillin and vanillin reduces contraction caused by L-type Ca2+ channel activation.

Round 2

Reviewer 1 Report

can be accepted in the present form